# Myelofibrosis and Survival Prognostic Models: A Journey between Past and Future

**DOI:** 10.3390/jcm12062188

**Published:** 2023-03-11

**Authors:** Andrea Duminuco, Antonella Nardo, Gaetano Giuffrida, Salvatore Leotta, Uros Markovic, Cesarina Giallongo, Daniele Tibullo, Alessandra Romano, Francesco Di Raimondo, Giuseppe A. Palumbo

**Affiliations:** 1Hematology Unit with BMT, A.O.U. Policlinico “G. Rodolico-San Marco”, Via S. Sofia 78, 95123 Catania, Italy; 2Dipartimento di Scienze Mediche, Chirurgiche e Tecnologie Avanzate “G.F. Ingrassia”, University of Catania, 95123 Catania, Italy; 3Dipartimento di Scienze Biomediche e Biotecnologiche, University of Catania, 95123 Catania, Italy; 4Dipartimento di Specialità Medico-Chirurgiche, CHIRMED, Sezione di Ematologia, University of Catania, 95123 Catania, Italy

**Keywords:** prognostic score, prognosis, myelofibrosis, overall survival, risk category

## Abstract

Among the myeloproliferative diseases, myelofibrosis is a widely heterogeneous entity characterized by a highly variable prognosis. In this context, several prognostic models have been proposed to categorize these patients appropriately. Identifying who deserves more invasive treatments, such as bone marrow transplantation, is a critical clinical need. Age, complete blood count (above all, hemoglobin value), constitutional symptoms, driver mutations, and blast cells have always represented the milestones of the leading models still used worldwide (IPSS, DIPSS, MYSEC-PM). Recently, the advent of new diagnostic techniques (among all, next-generation sequencing) and the extensive use of JAK inhibitor drugs have allowed the development and validation of new models (MIPSS-70 and version 2.0, GIPSS, RR6), which are continuously updated. Finally, the new frontier of artificial intelligence promises to build models capable of drawing an overall survival perspective for each patient. This review aims to collect and summarize the existing standard prognostic models in myelofibrosis and examine the setting where each of these finds its best application.

## 1. Introduction

Myelofibrosis (MF) is a rare and challenging type of bone marrow disorder that affects the ability to produce healthy blood cells. This myeloproliferative neoplasm (MPN) can arise de novo (primary myelofibrosis, PMF) or following another MPN (post polycythemia vera, PPV, or post essential thrombocythemia, PET). In both cases, the normal bone marrow is replaced by fibrous tissue, with the consequence of an impaired production of circulating blood cells. This impaired hematopoietic function can lead to the symptoms of MF, such as anemia, fatigue, night sweats, and weakness. Commonly, other manifestations are enlarged spleen and liver, abdominal discomfort, bleeding, and weight loss.

In brief, widely heterogeneous clinical manifestations characterize MF patients. Cytogenetic tests and CD34+ blast counts are routinely performed at the time of diagnosis and during follow-up to adequately study the patients [1,2]. From a molecular biology point of view, the role of driver mutations involving the *JAK*, *CALR*, and *MPL* genes, specific for uncontrolled cell proliferation, has been widely described. New analysis techniques, such as next-generation sequencing (NGS), play a key role in identifying mutations capable of conferring an increased risk of progression and worse outcomes [3].

Treatment options for myelofibrosis range from supportive care (i.e., transfusional support) to stem cell transplants (HSCTs), which, at the moment, are the only potentially curative treatment. Nevertheless, a wide range of drugs have recently seen the light and have significantly changed the clinical and therapeutic approach to this disease. In recent years, JAK inhibitors (JAKi) have played a leading role, with the first-in-class ruxolitinib paving the way for others (e.g., fedratinib, momelotinib, pacritinib) with potential to reduce symptoms, improve quality of life, reduce spleen size [4], and potentially improve survival.

However, despite the available treatments, we deal with a multifaceted disease characterized by a remarkably heterogeneous outcome. In fact, overall survival can range from fewer than two years to a life expectancy comparable to the general population and greater than 20 years [5]. Starting from these assumptions and considering the increasingly widespread therapeutic opportunities available, it is important to adequately stratify patients and be aware of how to direct them towards more or less intensive treatments. For this purpose, numerous prognostic scores have been proposed over the years and used to help physicians estimate median overall survival (mOS) and leukemia-free survival (LFS). At the end of the 1990s, several working groups proposed models to stratify patients. Hemoglobin < 10 g/dL, white blood cell count < 4 or >30 × 10^9^/L, absolute monocyte count > 1 × 10^9^/L, presence of constitutional symptoms, platelet count < 100 × 10^9^/L, and blast ≥ 1% were early prognostic hematological parameters on which the first models were based [6,7,8,9]. Then, many other models were developed, the most recent ones with the help of molecular biology and artificial intelligence (AI) [10].

The objective of this review is to collect all the currently available prognostic scores to explain in a clear, concise, and easily usable way which could be the most suitable model to apply at any time during patient follow-up, dwelling on the most recent proposals to have seen the light and that will probably be used shortly.

## 2. Prognostic Model Journey

### 2.1. IPSS (International Prognostic Scoring System)

One of the first prognostic scores created and validated in the primary MF setting is the so-called IPSS. A total of 1054 patients with a new diagnosis of MF referred in seven different centers in about 27 years of observation were included to develop the score [11]. The diagnosis was based on the criteria available at that historical moment and the long observation period must be considered.

At the presentation of the disease, the risk factors associated with short survival were selected through a stepwise Cox regression model. Age greater than 65 years, presence of constitutional symptoms, hemoglobin < 10 g/dL, WBC count > 25 × 10^9^/L, and blood blast > 1% obtained a *p*-value < 0.001, without marked difference among them in the hazard ratio (HR). By assigning one point for each variable, four risk classes were identified (low, intermediate-1 and -2, and high), characterized by a mOS between 135 and 27 months. In this large cohort of patients, during a median survival of 69 months, 49% had died. The most frequent causes of death were leukemic transformation, MF progression, thrombosis and cardiovascular complications, and infections. Compared to the general population, the survival during the first five years is superimposable in low risk patients, while the difference is evident after three years in intermediate-1 risk patients and already at diagnosis in intermediate-2/high risk patients. IPSS is a globally used prognostic model based on simple variables easily accessible in every center and commonly collected at diagnosis in the clinical practice, giving the opportunity of roughly foreseeing the patient’s outcome. However, it remains a less accurate model than the newer ones as it does not consider increasingly impacting molecular and genetic variables.

### 2.2. DIPSS (Dynamic IPSS) and DIPSS Plus

If IPSS was validated only for MF diagnosis, Passamonti et al. deepened the score by evaluating data able to accurately predict the OS and outcome of the patients during follow-up [12]. In total, 525 patients were stratified at baseline with the commonly used IPSS. Through a multivariable time-dependent Cox regression, they could assign a different weight to each of the five variables used in the IPSS based on HR. Age, WBC count, constitutional symptoms, and blast rate were characterized by a superimposable HR (1.98, 1.75, 2.06, and 1.86, respectively), and thus, a score of 1 was assigned. HR for anemia with hemoglobin < 10 g/dL was 4.18, assigning two points for this variable. Starting from these assumptions, the risk categories were defined as low (zero points), intermediate-1 (one or two), intermediate-2 (three or four), and high (five or six). Finally, an age-adjusted DIPSS (aaDIPSS) for patients under 65 years of age, and therefore eligible for HSCT, was implemented, removing the age variable. Then, the other variables used in DIPSS were confirmed as independent risk factors. The four risk categories identified for the general DIPSS were maintained, establishing five points as the maximum score. A Cox-survival regression model demonstrated the adequate potential of this model even in the setting of younger patients who could benefit from the transplant procedure. Death from any cause was reported for 53% of the patients, with disease evolution in 40%, followed by infections and coagulation disorders.

Therefore, the main strength of the DIPSS is its applicability at any time during the PMF clinical course, emphasizing how anemia is often the earlier risk factor acquired during follow-up, manifesting myelodepletion and representative of disease progression.

Finally, a further refined version of DIPSS was presented a few years later based on a more extensive clinical data set [13]. Univariate analysis showed that higher DIPSS scores (from intermediate-1 to high), unfavorable karyotype (defined as complex karyotype or sole or two abnormalities that include +8, −7/7q-, i(17q), −5/5q-, 12p-, inv(3), or 11q23 rearrangement), red blood cell transfusion dependency, and platelets lower than 100 × 10^9^/L were associated with lower OS. Through HR-weighted scoring, one point to each clinical variable and DIPSS intermediate-1, two points to DIPSS intermediate-2, and three points to the DIPSS high-risk class were assigned. This model, called DIPSS Plus, also established four risk classes: low risk (zero points; mOS of 180 months), intermediate-1 risk (one; mOS of 80 months), intermediate-2 risk (two to three; mOS of 35 months), and high risk (four to six; mOS of 16 months). Identifying potential prognostic factors for leukemic transformation is a plus of the score.

As well as IPSS, the variables considered in DIPSS are easily obtainable, improving the accuracy even during the course of the disease and underlining the key role of anemia. However, new scores that consider molecular biology could be more suitable, especially in patients potentially eligible for transplantation. The DIPSS Plus model tries to take a step beyond this boundary, but its limitation lies in the need for a karyotype, which is not always easily obtainable due to the frequent occurrence of a dry tap in bone marrow aspiration secondary to the pathognomonic fibrous scar tissue.

### 2.3. MIPSS70 (Mutation-Enhanced International Prognostic Score System) and MIPSS70+

The progressive discovery and the better knowledge of the driver mutations (in JAK2, CALR, and MPL) characterizing 85–90% of PMF patients, as well as the increasingly widespread availability of molecular biology study techniques, such as NGS [3], has allowed the development of prognostic models able to take into consideration the biology of the disease, which is known to be capable to impact its prognosis [14]. An international cooperative study enrolled a large number of PMF patients in whom the mutational panel of the most common myeloid genes were studied. *EZH2*, *ASXL1*, *IDH1/2*, and *SRSF2* mutations were found to confer a high risk (HRM) [15]. Two points were assigned in the case of two or more high-risk mutations, leukocytes > 25 × 10^9^/L, and platelets < 100 × 10^9^/L; one point to hemoglobin < 10 g/dL, blast > 2%, bone marrow fibrosis with grade ≥ 2, single HRM, and non-CALR type 1 presence. The patients were then stratified into three risk classes with different mOS.

The same working group has proposed a further implementation of this score, called MIPSS70+, where the cytogenetic alterations are weighted [16]. In this score, the maximum overall score is 12 points, as three points are added when an unfavorable karyotype (HR 3.1) is present. Four class risks were identified, improving the prognostic potential of this model. In both the training and validation cohorts, MIPSS70 and MIPSS70+ confirmed the potential to predict an inferior OS and a higher incidence of leukemic progression in high-risk patients (HR 14.4–9.1 to 22.7—and 13.3–4.7 to 37.5—respectively).

Here, for the first time, clinical, pathological, mutational, and cytogenetic parameters are incorporated and used to address most of the aspects that characterize MF heterogeneity. However, not all the necessary variables are always available in all patients, above all in elderly patients with follow-up in non-referral centers.

### 2.4. GIPSS (Genetically Inspired Prognostic Scoring System)

The importance of genetics has gradually been recognized in MF. In 2018, a score exclusively based on the patient’s genetic data was proposed [17]. In this score, called GIPSS, karyotype plays a leading role, assigning two points for very high-risk cytogenetics, one if the karyotype is unfavorable, and zero if favorable. Among the driver mutations, CALR plays a favorable role; thus, one point is assigned in case of absence. Finally, among the high-risk molecular mutations, one point is awarded in the presence of *ASXL1*, *SRSF2*, or *U2AF1 Q157* mutations. Based on the sum of these variables, four risk classes are identified, with an mOS ranging from 2.6 to 26.4 years. Through multivariate analysis, karyotype, *SRSF2* mutations, *ASXL1* mutations, platelet count <100 × 10^9^/L, and circulating blasts ≥ 2% were significantly related to a leukemic transformation. Thus, AML remained the leading cause of death in this cohort, occurring in 73 out of 380 patients (59%).

As well as MIPSS70, this score proves that genetic parameters could be enough to define the disease and its prognosis, but it also has the limitation of the difficulty of obtaining these data.

### 2.5. MYSEC-PM (Myelofibrosis Secondary to PV and ET-Prognostic Model)

PMF has been extensively studied and prognostically characterized over the years, while few studies focused only on MF secondary to essential thrombocythemia or polycythemia vera. Moreover, IPSS and DIPSS are less informative on secondary MF prognosis [18]. Thus, a dedicated score called MYSEC-PM has been developed and, at the moment, it is the only one currently available and applicable in this patient setting [19]. In this score, clinical and molecular data are adequately integrated.

A visual nomogram (http://www.mysec-pm.eu/, accessed on 10 March 2023) is currently used to identify which of the four risk categories a patient belongs to: low risk (mOS not reached), intermediate-1 risk (mOS of 9.3 years), intermediate-2 risk (mOS of 4.5), and high risk (mOS of 2.0 years), based on six different variables. A CALR-unmutated genotype, blasts rate ≥ 3%, and hemoglobin < 11 g/dL were assigned two points; platelet count < 150 × 10^9^/L and constitutional symptoms one point; and 0.15 per the patient’s years of age. Death occurred in 169 of 685 patients due to disease progression (38%), blast phase (32%), second malignancy (7%), and infection (9%). 

The score was independently validated and is now routinely used in daily practice [20,21,22].

### 2.6. MTSS (Myelofibrosis Transplant Scoring System)

As discussed, HSCT remains the only potentially curative therapeutic approach for patients with MF. It is the clinician’s task to identify those patients eligible for an intensive transplant procedure and adequately balance the risks related to the baseline hematological disease and those associated with HSCT, capable of exponentially increasing the transplant-related mortality (TRM). Among these, infections due to profound immunosuppression [23], graft-versus-host disease [24], and endothelial dysfunctions [25] are the most critical. On these bases, in 2019, Gagelmann et al. developed a prognostic score, namely MTSS, able to accurately predict 5-year OS and TRM, suggesting the potential final decision (“Go”, “Slow Go”, or “No Go”) [26].

Patients aged 70 or below with Intermediate-1 risk harboring an ASXL-1 mutation, Intermediate-2 risk, or high risk, based on DIPSS or MIPSS-70 models (depending on the availability of NGS data) for PMF and on MYSEC-PM for SMF, are to be considered for BMT upfront. Then, seven different variables related to the underlying disease (platelet and leukocyte values, driver mutation genotype, and presence of an *ASXL1* mutation), patient characteristics (age, Karnofsky performance status), and donor–recipient HLA matching grade are evaluated. One point is assigned to each of these characteristics, stratifying patients into four TRM risk classes (low, intermediate, high, and very high) and guiding the physician to advise for transplant.

### 2.7. RR6 (Response to Ruxolitinib after 6 Months)

Therapy with JAKi has primarily revolutionized the management of MF patients. Ruxolitinib (a first-in-class inhibitor of JAK1 and 2) is widely used mainly to control these patients’ disabling signs and symptoms (anemia, constitutional symptoms, splenomegaly) [27,28]. Several real-life data confirmed its efficacy and safety [20,29,30,31], even in elderly patients [32]. Despite this, in most patients, ruxolitinib must be stopped for toxicity or loss of response [33,34,35,36]. Thus, identifying the adequate timing for a possible treatment shift to a different JAKi or to HSCT for eligible patients is an open challenge. In this context, Maffioli et al. recently proposed a score able to identify “early” predictors of inferior survival after the first six months of therapy [37]. This model, called Response to Ruxolitinib after 6 months (RR6), was studied in a cohort of 209 patients and validated by different external groups [38,39]. RR6 divides the patients into risk classes considering the presence of a single of three variables (dose of ruxolitinib < 20 mg twice daily, palpable spleen length reduction from baseline ≤ 30%, red blood cell transfusion support at all times during the observation) at baseline, at three months, and at six months. The sum of these variables allows stratifying the patients as a low (< two points, median OS not reached), intermediate (between two and four points, with a median OS of 61 months), and high (> four points, median OS of 33 months) risk category. The strength of this prognostic model is that it helps to identify those who can benefit from an early change of therapy. An easily accessible tool is available online (http://www.rr6.eu/, accessed on 10 March 2023).

### 2.8. AIPSS-MF (Artificial Intelligence Prognostic Scoring System for Myelofibrosis)

Machine learning (ML) is a subfield of artificial intelligence that allows a computer system to create accurate predictions and make decisions built on extrapolations based on historical data without humans explicitly programming. ML potentially plays a role yet to be fully exploited in medicine, opening new scenarios. In particular, in the context of MF, registry data of 1386 patients followed up in 60 different Spanish centers were analyzed by this computational approach [10] to create a new score called AIPSS. The training set data modeled OS and leukemia-free survival (LFS) based on standard clinical features collected at diagnosis, establishing an individualized prediction for every single patient. The baseline parameters considered by this model are age, gender, presence or absence of constitutional symptoms, leukoerythroblastosis, hemoglobin, number of leukocytes and platelets, and peripheral blasts.

Despite the limitations due to the registry-based data, and thus their quality depending on the local physician entering statistics, the performance of the AIPSS model was assessed using cross-validated, time-dependent AUCs, confirming that it outperformed the gold-standard scores (IPSS and DIPSS). A further strong point is the possibility of creating a personalized risk built for each individual patient without the need to use genomic data that are not always easily accessible, especially in small centers or low-income countries.

A brief summary of all the above scores is reported in Table 1.

## 3. Discussion

Myelofibrosis is a very heterogeneous disease; its prognosis is not always easy to determine, especially on a single-patient basis. Numerous prognostic scores have been produced and presented in the literature, with a wide choice for the clinician to make the best decision to start or modify the treatment. This has become even more important recently as the therapeutical armamentarium has widened.

Nowadays, at diagnosis, the presence of driver mutations should be investigated in all patients, adding a routine NGS study in transplant-eligible individuals [40]. In fact, the genetic mutation landscape plays a critical role in the course of myelofibrosis. In the different scoring systems, the reported mutations able to generate a worse prognosis are different. However, mutations in *TP53*, *EZH2*, *CBL*, *U2AF1*, *SRSF2*, *IDH1/2*, *NRAS*, or *KRAS* are generally considered high-risk mutations [14]. The latest validated or ongoing developing prognostic models reserve an increasingly marginal role for constitutional symptoms, whose evaluation can be subjective, often leading to biases. On the contrary, the objective presence of specific mutations might determine the prognosis of MF. As an example, Luque Paz et al. divided their cohort of patients into four different genetic-based risk groups. In comparison with common prognostic models, this four-tier genomic classification combined with IPSS for PMF or MYSEC-PM for SMF achieved superiority versus MIPSS70 and its 2.0 version. Isolated *ASXL1*, instead, confirmed no definite role in prognostic outcome [14]. In the future, the discovery of other mutations to be included in the standard myeloid NGS panel will further improve the understanding of myelofibrosis biology and define single mutations or combinations that could radically change the clinical outcome.

In this context, the evaluation of allele burden might also play a role. In a study at MD Anderson, a high *JAK2 V617F* allele burden (with 50% as cut-off to dichotomize patients into two groups) represented a favorable marker of survival. Combined with an age lower than 65 years, the median OS was 126 months. Instead, those who carried a low allele burden reported an inferior OS, both in younger and elderly patients (72 and 35 months, respectively). The latter data was superimposable with that seen in triple-negative patients, known to have a worse prognosis [41].

Gene mutations are involved in the progression toward the blast phase. In all the developed models, its incidence is higher in patients categorized as high risk for OS, as expected. A study based on a large cohort of 1306 patients identified some mutations (in *IDH1*, *ASXL1*, and *SRSF2*), unfavorable karyotype, elderly age, male sex, rate of peripheral blood blasts ≥ 3%, moderate or severe anemia, and constitutional symptoms as variables correlated to a leukemic transformation within 5 years [42].

Prognostic scores should be applied both at the time of diagnosis and regularly during follow-up. In the event of a sudden change in clinical characteristics, the potential evolution of neoplastic subclones should be monitored, allowing an update of treatment if adverse genomic changes appear [43]. During PMF monitoring, the DIPSS and DIPSS Plus scores find greater indication where anemia plays a key role, since it is a progressive bone marrow failure. According to these prognostic scores and the indications of the National Comprehensive Cancer Network [44], it is a worldwide-accepted division in lower and higher MF risk, allowing patient stratification and guiding the physician through the treatment lines (Table 2). Younger patients at higher risk can be addressed with HSCT as the best therapy. In fact, once the eligibility transplant criteria are fulfilled and a good donor is available, the MTSS score suggests more benefits than risks.

Furthermore, considering the increasingly available therapeutic possibilities, a score built on the early response to the most used JAKi drug, i.e., ruxolitinib, has recently seen the light. Therefore, the RR6 score applied after 6 months of therapy represents a turning point in establishing an early shift to other treatments, sometimes radical, such as to HSCT. In this context, a validation of this score in the setting of patients eligible for transplantation could be helpful.

Looking at the future and considering the complexity of MPN diseases, we will move towards the construction of personalized prognostic models [45]. Artificial intelligence is gradually revolutionizing our approach to many fields of everyday human life. Specifically in medicine, AI could help generate desirable prognostic models targeting the single-patient level. In this context, AIPSS-MF represented the first milestone that, when validated on large external courts, will allow the exploration of new frontiers.

To sum up, the validation of more accurate models can gradually change the management of MF patients. At the time of diagnosis, the patients should be screened based on age and transplant eligibility. Second-level laboratory data (cytogenetics and myeloid mutational panel via NGS) should be performed in younger patients and based on these results, applying the MIPSS70, version 2.0 (preferred), or the GIPSS model, the patients with worse prognoses are identified. For elderly patients at diagnosis, the choice is IPSS for PMF and MYSEC-PM for SMF. During the course of the disease in PMF, DIPSS (or the Plus version, when karyotype is available) performs better than IPSS, giving more weight to anemia, while MYSEC-PM has been validated to be also used in SMF follow-up. MTSS and RR6 should be reserved for specific subgroups of patients, those treated with ruxolitinib and those to be transplanted, respectively. A possible accepted flowchart is reported in Figure 1.

## 4. Conclusions

Currently, several prognostic models can identify the patients with the worst outcome. The clinician’s challenge is to approach each patient in each phase of the disease carefully, extrapolating the results of the prognostic scores and thus obtaining critical information to guarantee the best therapeutic approach to best manage MF.

## Figures and Tables

**Figure 1 jcm-12-02188-f001:**
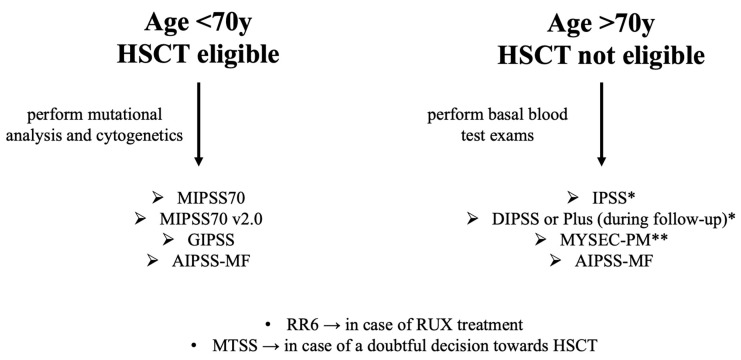
Flowchart based on age and HSCT eligibility for the choice of ideal prognostic models. HSCT: hematopoietic stem cell transplant; RUS: ruxolitinib. * used for PMF; ** used for secondary MF.

**Table 1 jcm-12-02188-t001:** Brief summary of all the prognostic models mentioned above that are actually used worldwide, with a demonstration of the explanation of the scores assigned to each variable and a description of the risk categories identified with their corresponding median overall survival. BM: bone marrow; HMR: high-molecular-risk; VHR: very-high-risk; RUX: ruxolitinib; RBC: red blood cells; NR: not-reached; OS: overall-survival.

Prognostic Model	Variable	Points	Risk Category	Score	Median Survival (Years)
**IPSS**	Constitutional symptoms	1	LOW	0	11.3
	Hemoglobin < 10 g/dL	1	INTERMEDIATE-1	1	7.9
	WBC > 25 × 10^9^/L	1	INTERMEDIATE-2	2	4
	Blast ≥ 1%	1	HIGH	≥3	2.3
	Age > 65	1	/	/	/
**DIPSS**	Constitutional symptoms	1	LOW	0	NR
	Hemoglobin < 10 g/dL	2	INTERMEDIATE-1	1–2	14.2
	WBC > 25 × 10^9^/L	1	INTERMEDIATE-2	3–4	4
	Blast ≥ 1%	1	HIGH	≥5	1.5
	Age > 65	1	/	/	/
**DIPSS-Plus**	DIPSS added to		LOW	0	15
	transfusion requirement	1	INTERMEDIATE-1	1	6.7
	Platelet count < 100 × 10^9^/L	1	INTERMEDIATE-2	2–3	2.9
	Unfavorable karyotype	1	HIGH	≥4	1.3
**MYSEC-PM**	Age	0.15 per year	LOW	<11	NR
	Hemoglobin < 11 g/d	2	INTERMEDIATE 1	≥11, <14	9.3
	Platelet count < 150 × 10^9^/L	1	INTERMEDIATE 2	≥14, <16	4.4
	Blasts > 3%	2	HIGH	≥16	2
	CALR-unmutated	2	/	/	/
	Constitutional symptoms	1	/	/	/
**MIPSS70**	Hemoglobin < 10 g/dL	1	LOW	0–1	27.7
	WBC > 25 × 10^9^/L	2	INTERMEDIATE	2–4	7.1
	Platelet count < 100 × 10^9^/L	2	HIGH	≥5	2.3
	Blasts ≥ 2%	1	/	/	/
	Constitutional symptoms	1	/	/	/
	BM fibrosis grade ≥ 2	1	/	/	/
	Single HMR mutation	1	/	/	/
	HMR mutations ≥ 2	2	/	/	/
	Non-*CALR* mutation type 1	1	/	/	/
**MIPSS70+** **v2.0**	Severe anemia	2	VERY LOW	0	NR
	Moderate anemia	1	LOW	1–2	16.4
	VHR karyotype	4	INTERMEDIATE	3–4	7.7
	Blasts ≥ 2%	1	HIGH	5–8	4.1
	Constitutional symptoms	2	VERY HIGH	≥9	1.8
	Unfavorable karyotype	3	/	/	/
	Single HMR mutation	2	/	/	/
	HMR mutations ≥ 2	3	/	/	/
	Non-*CALR* mutation type 1	2	/	/	/
**GIPSS**	Non-*CALR* mutation type 1	**1**	LOW	0	26.4
	VHR karyotype	**2**	INTERMEDIATE 1	1	8
	Unfavorable karyotype	**1**	INTERMEDIATE 2	2	4.2
	*ASXL1* mutation	**1**	HIGH	≥3	2
	*SRSF2* mutation	**1**	/	/	/
	*U2AF1 Q157* mutation	**1**	/	/	/
**MTSS**	HLA-mismatched donor	**2**	LOW	0–2	5 years-OS 83%
	Non *CALR*/*MPL* mutation	**2**	INTERMEDIATE 1	3–4	5 years-OS 64%
	Age > 57	**1**	INTERMEDIATE 2	5	5 years-OS 37%
	WBC > 25 × 10^9^/L	**1**	HIGH	>5	5 years-OS 22%
	Platelet count < 150 × 10^9^/L	**1**	/	/	/
	Karnofsky score < 90%	**1**	/	/	/
	*ASXL1* mutation	**1**	/	/	/
**RR6**	Spleen length reduction ≤ 30% with respect to baseline at 3 and 6 months	**1.5**	LOW	0	NR
	RBC transfusions requirement at baseline/3 months/6 months	**1 or 1.5**	INTERMEDIATE	1–2	5
	RUX treatment at dose < 20 mg twice daily at baseline, 3, and 6 months	**1**	HIGH	>2	2.7
**AIPSS-MF**	Sex, age (years), blood blasts (%), hemoglobin (g/L), leukocytes (×10^9^/L), platelet count (×10^9^/L), constitutional symptoms, leukoerythroblastosis		The model was designed to provide personalized predictions of overall survival and leukemia-free survival. Outcome predictions are from disease diagnosis.		

**Table 2 jcm-12-02188-t002:** Basal stratification of myelofibrotic patients, divided into lower and higher risk, according to commonly used prognostic models. * only used in MF secondary to ET and PV.

	Prognostic Model	Threshold
Lower Risk	MIPSS-70	≤3
	MIPSS-70+ Version 2.0	≤3
	DIPSS-Plus	≤1
	DIPSS	≤2
	MYSEC-PM *	<14
Higher Risk	MIPSS-70	>3
	MIPSS-70+ Version 2.0	>3
	DIPSS-Plus	>1
	DIPSS	>2
	MYSEC-PM *	>14

## Data Availability

Not applicable.

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
