# Peer review of "Myelofibrosis and Survival Prognostic Models: A Journey between Past and Future"

_jcm, 2023, doi:10.3390/jcm12062188_

Round 1

Reviewer 1 Report

jcm-2242546

Myelofibrosis and Survival Prognostic Models: A Journey between Past and Future.

The review article “Myelofibrosis and Survival Prognostic Models: A Journey between Past and Future (jcm-2242546)” by Duminuco A, et al. summarized several prognostic model for MF. This review article was organized well and comprehensive. Therefore, I considered that there were little issue. I have several comments to improve this review article.

1.      High risk MF are categorized into potential progression to acute leukemia and severe bone marrow disfunction with several organ disorder. In these prognostic model, high risk disease can always show more frequently progression for overt leukemia? Or are another cause of death, such as infection, pointed out in high risk diseases in some of them? The author should add the comments that cause of death was similar, especially overt leukemia, or not in each prognostic models.      

2.      High risk genetic mutations were poor predictors in several prognostic models. However, genetic mutations, which was identified as high risk in these prognostic models, were different. Why were genetic mutations different in these two models? The author could add several comments about that.

3.      The authors divided into low and high risks in table2. Is the high risk identified as candidate for allogenic transplantation if the patient has optimal donor and do not have severe complication?

Reviewer 2 Report

It is a very comprehensive and contemporary review of the prognostic systems used in management of myelofibrosis. 

Overall the article offers a very good understanding of various prognostic models used, their evolution with time, their strength and limitations. 

Apart from the models described there are other models ( subject to wider validation) that eliminate the 'constitutional symptoms' used in many of the earlier prognostic models, which often leads to bias as these symptoms can be very subjective. 

Also, some of the recent models seem to incorporate the allele burden, especially JAK2 allele burden, to further stratify these patients' risk. 

Reviewer 3 Report

This work summarize Prognostic Scoring Systems for patients with Myelofibrosis.

The authors specify all Prognostic Scoring Systems available today.

The main purpose of the study should be not only bring to our attention the list of Prognostic Scoring Systems but give clinical impact about the systems.

The authors should describe more about strengths and weaknesses of each system.

What a clinical impact of the new scores as Cytogenetical and   Molecular?  If this new scores change the management of the patients.

Which Scoring Systems is more appropriate and useful according to age (>60 vs <60) or symptoms as anemia, splenomegaly and constitutional symptoms

The authors should change introduction- easy bruising and frequent infections are not common symptoms of patients with Myelofibrosis.   Additionally, production of white blood cell and thrombocytes is not a major problem of patients with Myelofibrosis, most of them have Leukocytosis and some of them Thrombocytosis

Round 2

Reviewer 3 Report

authors have revised the manuscript accroding to the comments